# Molecular Detection of *bla*_TEM_ and *bla*_SHV_ Genes in ESBL-Producing *Acinetobacter baumannii* Isolated from Antarctic Soil

**DOI:** 10.3390/microorganisms13030482

**Published:** 2025-02-21

**Authors:** Clara Pazos, Miguel Gualoto, Tania Oña, Elizabeth Velarde, Karen Portilla, Santiago Cabrera-García, Carlos Banchón, Gabriela Dávila, Fernanda Hernández-Alomia, Carlos Bastidas-Caldes

**Affiliations:** 1Grupo de Investigaciones Antárticas (GIAN), Universidad Técnica del Norte (UTN), Av. 17 de Julio 5-21 y Gral. José María Córdova, Ibarra 100150, Ecuador; capazosm@utn.edu.ec (C.P.); teonia@utn.edu.ec (T.O.); develarde@utn.edu.ec (E.V.); kmportillac@utn.edu.ec (K.P.); jscabrera@utn.edu.ec (S.C.-G.); 2Scientific Advisory Committee of General Directorate of Maritime Interests and Foundation for Maritime, Riverine, Lacustrine, and Coastal Development (DIGEIM-FUNDEMAR), Quito 170403, Ecuador; mijail_kirochka62@outlook.com; 3Environmental Engineering, Escuela Superior Politécnica Agropecuaria de Manabí Manuel Félix López (ESPAM-MFL), Campus Politécnico, 1701518, El Limón, vía Calceta-El Morro, Calceta 130601, Ecuador; cbanchon@uagraria.edu.ec; 4Maestria en Ciencias de la Salud, Facultad de Medicina, Universidad de las Américas, Quito 170125, Ecuador; gabrieladavila797@gmail.com; 5Grupo de Investigación en Biodiversidad, Medio Ambiente y Salud (BIOMAS), Universidad de Las Américas, Quito 170125, Ecuador; fgha97@hotmail.com; 6Instituto Nacional de Biodiversidad (INABIO), Quito 170507, Ecuador; 7Facultad de Ingeniería y Ciencias Aplicadas, Biotecnología, Universidad de Las Américas, Quito 170125, Ecuador

**Keywords:** antibiotic resistance genes, cold environment, human-impacted environment

## Abstract

The phenomenon of antimicrobial resistance (AMR) in cold environments, exemplified by the Antarctic, calls into question the assumption that pristine ecosystems lack clinically significant resistance genes. This study examines the molecular basis of AMR in *Acinetobacter* spp. Isolated from Antarctic soil, focusing on the *bla*_TEM_ and *bla*_SHV_ genes associated with extended-spectrum beta-lactamase (ESBL) production; Soil samples were collected and processed to isolate Antarctic soil bacteria. Molecular detection was then conducted using polymerase chain reaction (PCR) to identify the bacteria species by 16S rRNA/*rpoB* and 10 different beta-lactamase-producing genes. PCR amplicons were sequenced to confirm gene identity and analyze genetic variability. *Acinetobacter baumannii* were identified by both microbiological and molecular tests. Notably, both the *bla*_TEM_ and *bla*_SHV_ genes encoding the enzymes responsible for resistance to penicillins and cephalosporins were identified, indicating the presence of resistance determinants in bacteria from extreme cold ecosystems. The nucleotide sequence analysis indicated the presence of conserved ARGs, which suggest stability and the potential for horizontal gene transfer within microbial communities. These findings emphasize that AMR is not confined to human-impacted environments but can emerge and persist in remote, cold habitats, potentially facilitated by natural reservoirs and global microbial dispersal. Understanding the presence and role of AMR in extreme environments provides insights into its global dissemination and supports the development of strategies to mitigate the spread of resistance genes in both environmental and clinical contexts.

## 1. Introduction

Antarctica, recognized as an ecological benchmark due to the pristine conditions safeguarded by the Protocol on Environmental Protection to the Antarctic Treaty [1], is increasingly threatened by human activities and climate change. These threats include air pollution, fuel spills, inadequate waste management, and wastewater discharges containing microplastics, human waste, cosmetics, detergents, bacteria, and antibiotics. Such impacts progressively disrupt Antarctic ecosystems, endangering their natural state [2,3].

While Antarctic bacteria have been extensively studied for their biotechnological potential, making them candidates for bioremediation in cold environments [4], there is growing concern regarding the spread of antibiotic resistance genes (ARGs). These genes are likely introduced into Antarctic ecosystems through anthropogenic activities, including untreated wastewater and other human-driven contaminants. Among the most concerning anthropogenic pollutants is the introduction of human-associated bacteria, such as *Escherichia coli*, *Salmonella enterica*, *Acinetobacter baumannii,* and *Legionella* spp., through untreated wastewater disposal into these fragile environments [5].

The genus *Acinetobacter* has drawn significant attention to its ability to produce extended-spectrum beta-lactamases (ESBLs), which are a striking example of enzyme-mediated bacterial resistance. These enzymes, known for conferring resistance to multiple antibiotics, including penicillin, cephalosporins, and monobactams, have been detected not only in human-associated environments but also in seemingly unexposed habitats such as Antarctic soils and glaciers [6].

The acquisition of multidrug resistance genes, such as beta-lactamase and/or carbapenem resistance genes, is still a major clinical concern, which can be disseminated through various HGT mechanisms reinforced by anthropogenic and environmental phenomena [7]. While the majority of research has concentrated on *A. baumannii*, a well-established nosocomial pathogen and reservoir of ARGs [8,9,10,11], it is responsible for various infections, particularly ventilator-associated pneumonia and bloodstream infections [12]. Notably, the *Acinetobacter* genus exhibits remarkable genetic plasticity, rapidly acquiring resistance determinants through horizontal gene transfer (HGT) mechanisms [13]. However, the contribution of intrinsic antibiotic resistance (IAR) to the environmental dissemination of *A. baumannii* has been poorly studied.

In this context, antimicrobial resistance (AMR) in the *Acinetobacter* genus remains a global challenge, affecting human health, agriculture, and the environment. According to The Lancet, 1.27 million deaths were attributable to drug-resistant infections in 2019, a number projected to rise to 10 million annually by 2050 [14]. While much of the focus on AMR has been on its impact on clinical settings, the broader implications for environmental systems are becoming increasingly apparent. This raises critical questions about the extent of AMR’s influence in ecosystems like Antarctica and the methodologies required to uncover its full magnitude.

Human activities, including scientific research and tourism, significantly contribute to the dissemination of ESBL genes such as *bla*_CTX-M-1_ and *bla*_CTX-M-15_ in polar regions. Furthermore, migratory birds further amplify this issue by acting as vectors for resistant bacteria, connecting distant ecosystems, and facilitating the global spread of AMR [15]. As a result, the Antarctic coastal microbial communities, particularly those exposed to wastewater discharge and rainfall runoff, experience increasing levels of antibiotic exposure and ARG proliferation where the HTG plays a critical role, spreading ARGs across microbial populations in these ecosystems [16].

In addition to the well-documented acquisition of ARGs through HGT, *A. baumannii*. inherently possess genome-encoded resistance mechanisms, underscoring the dual origin of resistance within microbial populations [17]. This duality is particularly relevant in ecosystems like Antarctica, where both innate and acquired resistance may converge, shaping the adaptive potential of this bacteria in response to environmental and anthropogenic pressures.

Given this backdrop, the Protocol on Environmental Protection to the Antarctic Treaty underscores the critical importance of fostering international collaboration in scientific research, thereby promoting the exchange of data and findings to address these pressing challenges. Additionally, nations conducting research in the Antarctic bear a responsibility to comply with rigorous environmental and waste management standards to mitigate their ecological impact. This study builds upon these principles by employing molecular tools to identify and characterize ESBL genes in *Acinetobacter* spp. isolated from Antarctic soil. Through the detection of this ARG, this research seeks to illuminate the mechanisms underpinning antibiotic resistance in extreme environments, explore anthropogenic contributions to its emergence, and advocate for robust mitigation strategies to preserve Antarctica’s ecological integrity.

## 2. Materials and Methods

### 2.1. Area of Study and Sampling

The South Shetland Islands, situated off the northern tip of the Antarctic Peninsula, constitute a key part of this unique polar region, serving as a focal point for scientific research conducted by numerous countries. Among these islands, Greenwich Island hosts the Ecuadorian Research Station Pedro Vicente Maldonado (−62.449267, −59.742269).

The region is characterized by a polar maritime climate, with mean annual temperatures ranging from −2 °C to 1 °C, elevated humidity, and frequent precipitation, predominantly as snow and occasional rain. The island’s terrain is marked by rocky outcrops, glacial formations, and sporadic patches of ice-free soil, which support a limited but ecologically significant vegetation, primarily consisting of mosses, lichens, and microbial communities adapted to the harsh environmental conditions.

A kilogram of soil was collected at a depth of 15 cm near the Pedro Vicente Maldonado Research Station (Figure 1). The sample was obtained using a sterile spoon to prevent contamination and processed using the soil quartering method to ensure the representativeness of the heterogeneous substrate. Following collection, the sample was preserved under a controlled cold chain at 4 °C to maintain microbial viability and transported to the laboratories of the Technical University of the North in Ecuador for further analysis.

### 2.2. Bacteria Isolation and Identification

Serial dilutions of the soil suspension were prepared, and 0.1 mL from each dilution was plated on Luria–Bertani (LB) agar (Merck, Burlington, MA, USA) and incubated at 4 °C [18]. A pure culture was obtained from the initial plating by successive quadrant streaking [19,20]. After four rounds of purification on LB agar at 4 °C, the strain developed characteristic round and reddish colonies.

Gram staining was performed from the colonies obtained, and Gram-negative colonies were subsequently cultured on a selective MacConkey agar (Merck, Darmstadt, Germany). Lactose-positive and lactose-negative colonies were further inoculated onto CHROMagar™ Acinetobacter, ECC (*E. coli*/Coliforms) agar (both from CHROMagar, Paris, France), and cetrimide agar (Merck, Darmstadt, Germany) to facilitate the identification of specific bacteria. Suspected colonies of any culture were isolated and identified using the API 20NE biochemical identification system (bioMérieux, Marcy-l’Étoile, France), which provided a comprehensive metabolic profile for the isolates. The isolates were preserved in a brain heart infusion broth with 10% glycerol (Becton Dickinson, Sparks, MD, USA). Isolates used in this study were stored at −20 °C, while those preserved for future research were frozen at −80 °C.

### 2.3. Molecular Characterization of the Isolated Strain

Genomic DNA was extracted from pure culture by resuspending two colonies in 200 µL of TE (10 mM Tris–HCl and 1 mM EDTA) in a 1.5 mL microtube. Afterwards, the tubes were placed at −20 °C overnight. At the end of the incubation, the samples were sonicated in a water bath for 30 min at 4 °C in an Ultrasonic Bath 5.7 L CPX3800 Fisherbrand (Fisher Scientific, Waltham, MA, USA). After 30 min of sonication, the tubes were placed on ice for 10 min, heated at 95 °C for 5 min, refrigerated again at −20 °C for 5 min, and vortexed three times. Finally, the tubes were centrifuged at 14,000× *g* for 1 min, and the supernatant was transferred to a new 1.5 microtube to subsequently quantify the DNA using a NanoDrop 2000 Spectrophotometer from Thermo Scientific (Waltham, MA, USA) and adjust it to a concentration of 60 ng/µL.

The DNA was used to perform an endpoint polymerase chain reaction (PCR) using the GoTaq^®^ Green Master Mix kit (Promega, Madison, WI, USA), with a final volume of 15 µL, with the following concentrations: primer forward and reverse at 0.4 µM, genomic DNA at 4 ng/µL, and GoTaq^®^ Green Master Mix at 1X. For identification, the primers 8F (5′-AGAGTTTGATCCTGGCTCAG-3′) and 534R (5′-ATTACCGCGGCTGCTGG-3′) directed to ribosomal RNA 16S gene were used [21]; an *Escherichia coli* strain (ATCC 25922) was utilized as a positive control. Additionally, the primers rpoBF (5′-GGCGAAATGGCWGAGAACCA-3′) and rpoBR (5′-GAGTCTTCGAAGTTGTAACC-3′) were used for the amplification of the 1074 bp region of the beta-subunit of bacterial RNA polymerase (*rpoB*) [22].

The thermocycling conditions for the 16S rRNA gene consisted of an initial denaturation at 95 °C for 5 min, then 35 cycles of denaturation at 95 °C for 30 s, annealing at 55 °C for 30 s, extension at 72 °C for 30 s, and a final extension of 5 min at 72 °C. For the *rpoB* gene, the annealing temperature was 50 °C. The PCR product was visualized by 2% agarose gel electrophoresis.

### 2.4. Molecular Identification of Betalactamase-Encoding Genes

For the detection of ESBL genes, a single-endpoint PCR was conducted targeting six ESBL genes: *bla*_TEM_, *bla_SHV_*, *bla_CTX-M-1_*, *bla_CTX-M-2_*, *bla_CTX-M-8/25_*, and *bla_CTX-M-9_*, following the primer design described by Hoa et al. [23]. Additionally, single-endpoint PCR was employed for the identification of carbapenemase genes (*bla*_OXA48,_
*bla*_KPC_, *bla*_NDM_, and *bla*_VIM_) based on previously established protocols [24]. The primer sequences and the annealing temperatures are presented in Appendix A.

The PCR products were resolved on a 2% (*w*/*w*) agarose gel stained with SYBR™ Safe (Invitrogen, Waltham, MA, USA) and prepared with 1X TBE buffer. Electrophoresis was performed at 100 V for 35 min using a Labnet Enduro Gel XL horizontal chamber (Labnet International, Inc., Edison, NJ, USA). Visualization of the agarose gels was carried out using a ChemiDoc™ Imaging System (BioRad, Hercules, CA, USA), with analysis facilitated by Image-Lab™ software (BioRad, USA, version 6.1). The amplicon lengths were determined by comparison against a 100 bp DNA ladder (Invitrogen, USA). DNA-positive controls for the target genes were used from a previous study [25], while a non-ESBL-producing *A. baumannii* strain (ATCC BAA-1605) was utilized as a negative control.

### 2.5. Sequencing Identification and Phylogenetic Analysis

The 16S rRNA gene and *rpoB* amplicons were sequenced by Sanger system in an ABI 3500xL Genetic Analyzer sequence (Applied Biosystems, Waltham, MA, USA). The resulting sequences were identified using NCBI BLAST (accessed on 31 January 2025; https://blast.ncbi.nlm.nih.gov/Blast.cgi). A maximum likelihood phylogenetic tree was constructed in NGPhylogeny software (accessed on 31 January 2025; https://ngphylogeny.fr) [26] with the following model: Tamura-Nei 93 (TN93) + gamma distribution (G) + invariable sites (I) for the 16S rRNA tree, and General Time Reversible (GTR) + G + I for the *rpoB* and concatenated tree. *Klebsiella pneumoniae* was used as an outgroup. The tree was constructed with 500 bootstraps. The sequences were uploaded to GenBank under the accession numbers PV054918 (16S rRNA gene) and PV059848 (*rpoB*).

### 2.6. Detection of Plasmid Incompatibility Groups

Plasmid detection in the strain was performed using a PCR-based replicon typing method to identify incompatibility (Inc) groups, including HI1, HI2, I1-ly, X, L/M, N, FIA, FIB, W, Y, P, FIC, A/C, T, FIIAs, F, K, and B/O [27]. The PCR conditions for all incompatibility group, with exception for IncF were performed as follows: initial denaturation at 94 °C for 5 min, then 30 cycles of denaturation at 94 °C for 1 min, annealing at 60 °C for 30 s, extension at 72 °C for 1 min, and a final extension at 72 °C for 5 min. For Inc F PCR, an annealing temperature of 52 °C was used.

## 3. Results

Pure colonies were successfully obtained from Antarctic soil samples, initially identified as belonging to the *Acinetobacter* genus through traditional microbiological tests. The biochemical analysis using the API 20NE system yielded conflicting results regarding the strain identification, with some metabolic and enzymatic tests aligning with *A. johnsonii* while others were consistent with *A. baumannii*. The glucose oxidation test (GLU) was positive, confirming the strain’s ability to oxidize glucose under aerobic conditions, a characteristic of non-fermenting bacteria. The nitrate reduction test (NO_3_) was negative, which is typical for *A. johnsonii*, whereas the arginine dihydrolase test (ADH) was positive, aligning with the metabolic profile of *A. baumannii*.

Additional enzymatic tests further reflected this discrepancy. The urease test (URE) was negative, as expected for both species, while gelatinase activity (GEL) was also negative, excluding gelatinase production. However, substrate utilization tests showed variability: while the oxidation of mannitol (MAN) and arabinose (ARA) were negative—consistent with *A. baumannii*—the gluconate oxidation (GLU) test showed weak positive activity, which is more characteristic of *A. johnsonii*.

The PCR amplification successfully yielded amplified conserved regions (amplicons) of approximately 650 and 1074 base pairs, consistent with the expected size for 16S rRNA gene and *rpoB* gene fragments, respectively (Appendix A). The BLAST analysis, excluding uncultured and environmental sample sequences, confirmed that the isolate matched *A*. *baumannii* with 100% identity and 100% query coverage in both genes. Phylogenetic analysis based on individual and concatenated 16S rRNA gene (Appendix A) and *rpoB* (Appendix A) sequences further confirmed the taxonomic assignment of the microorganism within the *A. baumannii* clade (Figure 2).

The screening of AMR genes shows the presence of two ESBL-encoding genes, *bla*_TEM_ and *bla*_SHV_, amplicon sizes of 372 and 231 base pairs, respectively (Appendix A). No carbapenemase genes or incompatibility (Inc) plasmid groups were detected in the strain.

## 4. Discussion

To our knowledge, this is the first report of an ESBL-producing *A. baumannii* strain isolated from Antarctic soil. *Acinetobacter baumannii* can readily adapt to cold environments such as Antarctica through a process known as homeoviscous adaptation [28]. Despite being primarily classified as a nosocomial pathogen, *A. baumannii* exhibits remarkable bioremediation potential. A previous study reported a 91% efficiency in degrading textile industry wastewater dyes [29]. Moreover, *A. baumannii* has demonstrated the ability to degrade hydrocarbons, such as diesel, particularly in mixed-culture environments [30,31]. Additionally, it has been employed in bioemulsifier and biosurfactant production, as well as in the degradation of polycyclic aromatic hydrocarbons [32].

Despite their ecological and biotechnological potential, *Acinetobacter* species are also recognized as reservoirs for antibiotic resistance genes (ARGs) in Antarctic ecosystems. The first reports of antibiotic-resistant *Acinetobacter* from water samples in Antarctica date back to 2013 [33,34]. More recently, *A. radioresistens*, isolated from Antarctic soil, was found to harbor multiple ARGs, including *bla*_OXA-23-like_ and *bla*_PER-2_, genes typically associated with nosocomial settings [35]. These findings raise concerns about the spread of clinically relevant ARGs into even the most remote environments [34].

The observed discrepancy between phenotypic and molecular identification can be attributed to the inherent limitations of biochemical assays, particularly when analyzing environmental strains isolated from extreme habitats such as Antarctic soil [36,37]. In this environment, bacteria are subjected to unique selective pressures that may drive metabolic adaptations not accounted for in conventional clinical databases [38]. The API 20NE system relies on the phenotypic expression of metabolic traits under standardized laboratory conditions, which may not be sufficient for identifying strains exhibiting atypical biochemical profiles, as is often the case with microorganisms from polar regions.

The detection of *bla*_TEM_ and *bla*_SHV_ among the ten antibiotic resistance genes (ARGs) tested in *A. baumannii* underscores the presence of an *Acinetobacter* species typically associated with nosocomial environments and highlights the dissemination of ESBL genes in Antarctic soil. These findings are significant as they suggest the potential dissemination of clinically relevant ARGs even in remote and pristine environments. Recent studies have demonstrated the existence of antimicrobial resistance (AMR) in Antarctic bacteria, underscoring their capacity to resist a multitude of antibiotics even in isolated ecosystems. Genes such as *bla*_TEM_, *bla*_SHV_, and *bla*_CTX-M_ have been identified in enterobacteria, as well as in Antarctic strains with antimicrobial potential [2]. The research conducted in soil and seawater in the vicinity of Antarctic bases indicates that human activities have contributed to the dissemination of resistant genes [5,6].

Identifying these ARGs in *A. baumanni* from Antarctic soil is particularly noteworthy as these genes have historically been associated with resistance to penicillin and early-generation cephalosporins, and their presence may indicate anthropogenic influence. Previous studies have documented the dissemination of these genes through horizontal gene transfer mediated by mobile genetic elements (MGEs) such as plasmids and transposons [13,39]. *Acinetobacter baumannii* are particularly well-known for their exceptional capacity for HGT, which enables them to acquire and integrate genetic material from diverse sources [40]. This adaptability enhances their survival in challenging environments and facilitates the persistence and spread of ARGs across microbial communities. This raises questions about the role of human activity, including research station waste management and the global migration of resistance genes, introducing and maintaining these ARGs in polar environments [33].

Interestingly, no carbapenemase genes (*bla*_OXA-48_, *bla*_KPC_, *bla*_NDM_, or *bla*_VIM_) were detected in this study, nor were other ESBL genes, such as the *bla*_CTX-M_ group, which are highly prevalent and frequently reported in South America [41]. This aligns with previous studies suggesting that ESBL genes, particularly *bla*_TEM_ and *bla*_SHV_, may be among the first resistance determinants to spread into non-clinical settings, possibly due to their earlier emergence and global dissemination.

No incompatibility (Inc) groups were detected in the analysis, which suggests several possible explanations. First, the resistance genes may be located on plasmids outside the 18 Inc groups assessed, suggesting the involvement of rare or novel replicons [42]. Second, these genes might be integrated into the bacterial chromosome, a well-documented phenomenon in *A. baumannii* facilitated by mobile genetic elements such as transposons or integrons. This chromosomal integration not only stabilizes resistance traits but also underscores the remarkable genetic plasticity of this genus [43].

While continuous and active surveillance is needed to fully understand the presence and dissemination of ARGs in Antarctic microbial communities, this absence could reflect the unique selective pressures in these environments, which are markedly different from clinical and agricultural settings where such genes are commonly found [44]. The limited antibiotic exposure in Antarctic ecosystems, coupled with the pristine nature of the region, may explain the lower prevalence of these resistance determinants. However, the possibility that these genes exist at undetectable levels or within non-target bacterial populations cannot be ruled out. These findings highlight the importance of sustained monitoring efforts to track the emergence and spread of antibiotic resistance in extreme environments.

## 5. Conclusions

*Acinetobacter baumannii* isolated from Antarctic soil could play a crucial role as reservoirs of antibiotic resistance genes and deserve attention. The presence of clinically significant ARGs, including *bla*_TEM_ and *bla*_SHV_, underscores the need for careful monitoring and management of microbial communities in polar environments. These findings highlight the dual impact of *Acinetobacter* as a valuable ecological asset and as a potential contributor to the global spread of antibiotic resistance, even in the most pristine ecosystems. Further research is essential to unravel the mechanisms driving ARG dissemination in these unique habitats and to mitigate potential risks associated with anthropogenic activities.

### Study Limitations

This study has certain limitations that should be considered. First, the use of the 16S rRNA gene, while widely recognized for its utility in bacterial taxonomic identification and sufficient for the objectives of this research, has limited resolution for species-level differentiation. Future studies would benefit from the inclusion of additional molecular markers or advanced techniques such as whole-genome sequencing (WGS), which could provide more detailed taxonomic resolution and comprehensive insights into genetic content, including mobile genetic elements and ARGs [45].

Second, the transportation of samples from Antarctica to the laboratory presents challenges. Although the Ecuadorian Antarctic research station is well-equipped for initial sample handling and preservation, prolonged conservation and transportation times may impact methodologies sensitive to sample quality, such as DNA or RNA extraction. This could affect the viability of microorganisms or the integrity of nucleic acids, potentially influencing the results.

Third, while this study successfully detected and characterized the presence of ESBL-related genes, it was limited to molecular analyses and did not assess phenotypic antibiotic resistance. The presence of ARGs does not necessarily correlate with functional resistance, as gene expression and regulatory mechanisms can vary. The inclusion of phenotypic tests, such as disc diffusion assays or minimum inhibitory concentration (MIC) determinations, would provide critical insights into the actual resistance profiles of the isolates.

Despite these challenges, the traditional techniques used in this study proved adequate for the intended purpose, enabling the successful identification of *Acinetobacter* sp. and the detection of ARGs. Future research could benefit from on-site advanced methodologies or improved preservation strategies to mitigate the potential impact of transportation on sample integrity, particularly for studies requiring high-sensitivity analyses.

## Figures and Tables

**Figure 1 microorganisms-13-00482-f001:**
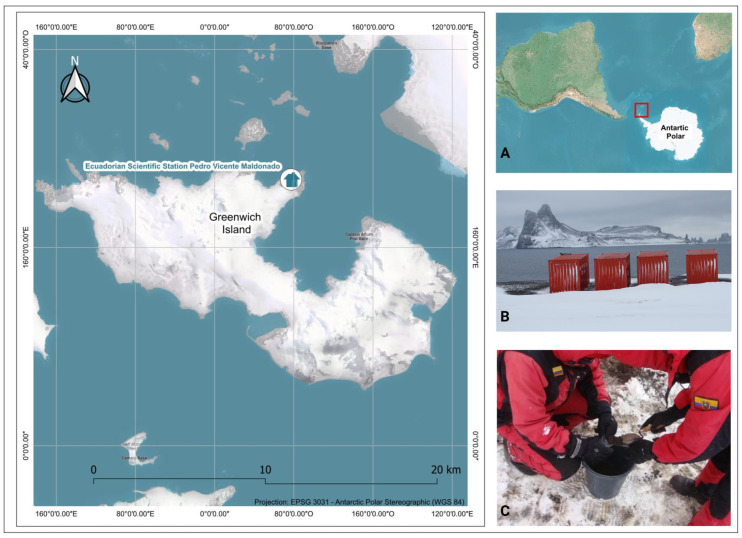
Location of the Ecuadorian scientific station Pedro Vicente Maldonado on Greenwich Island, Antarctica. (**A**) The geographic context of the Antarctic Peninsula highlights in red the location of Greenwich Island, which corresponds to the sampling site. (**B**) Station equipment and a close-up view showcasing the environmental conditions characteristic of the polar region. (**C**) Soil sampling conducted by the Ecuadorian scientific team near the station. The main map displays the precise location of the station on Greenwich Island, projected in the EPSG 3031 system (Antarctic Polar Stereographic).

**Figure 2 microorganisms-13-00482-f002:**
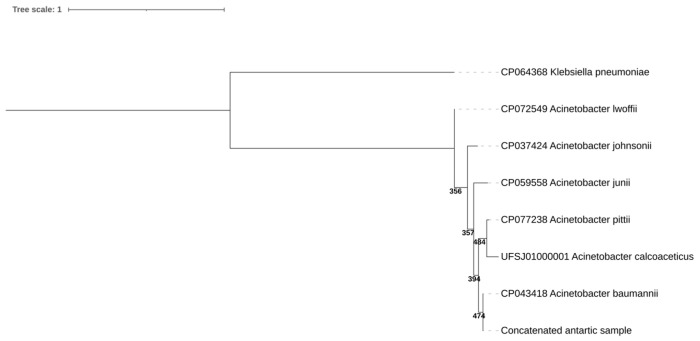
Concatenated (*rpoB*-16SrRNA) phylogenetic tree construction using the maximum likelihood method using the GTR + G + I model in NGPhylogeny software with 500 bootstraps. The bootstrap value is presented in the tree nodes.

## Data Availability

The original contributions presented in this study are included in the article/Appendix A. Further inquiries can be directed to the corresponding author.

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
