# Peer review of "Molecular Detection of blaTEM and blaSHV Genes in ESBL-Producing Acinetobacter baumannii Isolated from Antarctic Soil"

_microorganisms, 2025, doi:10.3390/microorganisms13030482_

Round 1
Reviewer 1 Report (Previous Reviewer 1)
Comments and Suggestions for Authors
The manuscript was very carefully corrected with considering all lost all my comments. The only one suggestion I have is providing bacterial name in Italic in Figure 2. However, it is of minor importance and probably could be manage during the paper production. Congratulation on the paper.
Author Response
Please see the attachment

Reviewer 2 Report (New Reviewer)
Comments and Suggestions for Authors
The Authors have isolated a Acinetobacter baumannii strain from the environment near their camp in Antarctica and molecularly detected it for the presence of antimicrobial resistant genes. The authors suggest that their presence is likely due to human influence. The manuscript overall is complete and the comments that I have are minor.
1) First, title words are key words so please think about changing the repeated words.
2) The authors sequenced rDNA and not rRNA so there are several places that need to be adjusted. For example, supplemental figure 2 needs the rRNA changed to rDNA. This needs to be completed throughout. It is ok to state 16S rRNA gene, but it should not be referred to as 16s rRNA as it would suggest the authors were sequencing RNA and not DNA.
3) Most publications should now be stand-a-lone. This means that there is one method that the authors only reference a previous work. Please consider adding a short "In brief" section that quickly covers all the major steps.
4) Lastly, natural population of bacteria have these genes to protect against some fungi and other soil antagonist. This could also be an explanation, particularly since the authors talk about the unique attributes of cold tolerant bacteria. I think adding just an acknowledgement of this possibility to the discussion would be also beneficial, but not necessary.
All comments are in the attached file

Author Response
Please see the attachment

Reviewer 3 Report (New Reviewer)
Comments and Suggestions for Authors
The authors have improved the manuscript accordingly. The abstract perfectly illustrates the conducted work; The Introduction provides the needed background to justify the need for performing the research; The methodologies are reasonably described so other investigators can reproduce the same experiments worldwide; The Results are also well presented and discussed with proper literature. My concerns are regarding the Conclusions, I believe the authors should elaborate on their conclusions in a separate section and indicate the practical implications of the obtained results as well as provide more directions for further studies.
Author Response
Please see the attachment

This manuscript is a resubmission of an earlier submission. The following is a list of the peer review reports and author responses from that submission.
Round 1
Reviewer 1 Report
Comments and Suggestions for Authors
Nowadays, the spread of antibiotic resistant genes is one of the biggest challenges from both medical and economic points of view. Thus, the study presented in the manuscript entitled ‘Molecular detection of blaTEM and blaSHV genes in ESBL-Producing Acinetobacter sp. isolated from Antarctic soil’ prepared by Pazos and co-workers and submitted to Microorganisms as a Communication, is quite essential, especially since it concerns bacteria associated with Antarctica. Nevertheless, before publication the Authors should modify the manuscript. I raise the following most important points, which may significantly increase the quality of the paper.
1. The part of Abstract named Conclusion should be divided into (i) Results and (ii) Conclusion as follows:
RESULTS: BACTERIA REPRESENTING the Acinetobacter genus WERE [was <-REMOVE] identified by both microbiologiCAL and molecular tests. Notably, both the blaTEM and blaSHV genes ENCODING THE ENZYMES RESPONSIBLE FOR [HERE PROVIDE THE ANTIBIOTIC NAMES] were identified, indicating the presence of resistance determinants in BACTERIA FROM extreme cold ecosystems. The NUCLEOTIDE sequence analysis indicated the presence of conserved ARGs, which suggest stability and the potential for horizontal gene transfer within microbial communities. CONCLUSIONS: These findings emphasize that AMR is not confined to human-impacted environments but can emerge and persist in remote, cold habitats, potentially facilitated by natural reservoirs and global microbial dispersal. Understanding THE PRESENCE AND ROLE OF AMR in extreme environments provides insights into its global dissemination and supports the development of strategies to mitigate the spread of resistance genes in both environmental and clinical contexts.
2. Keywords should be given in different order, e.g. Antarctic soil, Acinetobacter SPP. ANTIBIOTIC resistance genes, Extended-spectrum Beta-lactamases.
3. I fully agree that bacteria can acquire the antibiotic resistant genes, but also bacteria possess innate resistance. This issue should be at least mentioned in the Introduction (e.g. after the sentence ending in L. 48).
4. The bacteria isolation and the rest of the experiments are well planned and executed. However, I strongly recommend assessing the ESBL antibiotic resistance, e.g. using paper discs soaked with antibiotics. The presence of antibiotic resistance genes in bacteria does not mean the bacteria are antibiotic resistant. It would be of added value to test the antibiotic resistance, not only to test the gene presence in the isolates.
5. L. 124-125: please modified the sentence ‘Serial dilutions of the soil suspension were prepared, and 0.1 mL from each dilution 124 was plated on Luria Bertani (LB) agar (Merck, USA) and incubated at 4°C [17].’ as follows: ‘Serial dilutions of the soil suspension were prepared, and 0.1 mL from each dilution was plated on Luria Bertani (LB) agar (Merck, ADD TOWN, USA) and BACTERIA WERE incubated at 4°C [17].’
6. Chapter 2.2. Bacteria isolation and identification:
- Please add names of towns for suppliers of the reagents and material used in the study, e.g. Promega, TOWN?, USA.
- Please provide in the section 2.3. the size of the 16S rRNA gene PCR product.
- I suppose the Authors also considered the negative control for PCR reagents. I mean the author included in PCR the reaction without any bacterial DNA added.
7. Results and Discussion:
- I do not see the results from the API 20E tests. Please, describe the results in the main text and provide the outcomes in a Table in the Supplementary Material.
- Please provide PCR results for the antibiotic resistant genes
- Please correct the figure 2 by adding the PCR product's size and providing the bacteria names in Italic (panel A). Instead of 'simple' write the isolate number. Moreover, in the caption add ‘BASED ON THE 16S rRNA GENE SEQUENCE’ after ‘…. the isolated strains.’
8. Minor suggestions:
- L. 45: add a dot at the end of the sentence;
- L. 82: change HTG into HGT;
- L. 202: change ‘sample’ into ‘bacteria’;
- L. 223 and in the entire text: ‘sp.’ should be written using Italic fonts.
Reviewer 2 Report
Comments and Suggestions for Authors
Overall opinion
1. The experimental content of this paper is small, and the data of only one picture is not convincing, so it is suggested to supplement some experiments (such as Conjugative transfer, whole-genome sequencing etc.) to improve the data.
2. Only one picture was shown in the PCR results of articles 2.3 and 2.4. Should there be a picture for each of them?
3. Overall, the structure of the article is more discursive, the text content is too much experiment, too little data content, it is suggested to change.
4. Whether the two experiments involved in the article are representative? If yes, please list relevant experimental data or pictures; if no new experiments related to drug resistance are suggested for verification.
Specific suggestion
1. Line 30, the "suggest" should be changed to "suggests".
2. Line 45, 184, 215, the "." should be added at the end of the sentence.
3. Line 56, a reference should be added.
4. Line 58, 226, 239, the "environments" be changed to "environment", It is an uncountable noun.
5. Line 74, the "in" should be changed to "on".
6. Line 79, the "As result" should be changed to "As a result".
7. Line 165, the "et al" should be changed to "et al. ".
8. Line 182, the reference should be moved to the end of the sentence.
9. Line 189, the "is" should be changed to "was ".
10. Line 395, the spacing between words is too big.
Line, 408, the year should be shown in bold.
Comments on the Quality of English LanguageIf the quality of English language is further improved, it will be beneficial to the understanding of this study.
